

# Knee replacement outcome predicted by physiotherapists: a prospective cohort study

Marius Henriksen[1], Hiwa Mukriyani[1] and Carsten Juhl[2,3]

[1] The Parker Institute, Copenhagen University Hospital Bispebjerg-Frederiksberg, Copenhagen, Denmark
[2] Department of Physiotherapy and Occupational Therapy, Copenhagen University Hospital, Herlev-Gentofte, Copenhagen, Denmark
[3] Department of Sports Science and Clinical Biomechanics, University of Southern Denmark, Odense, Denmark

## ABSTRACT

**Background**. Knee arthroplasty (KA) is commonly used for osteoarthritis of the knee joint and it is a highly successful procedure. Still, KA leaves 20% of patients dissatisfied with their outcome. The purpose of this study was to determine if a prognosis made by physiotherapists at the orthopaedic wards during the first post-operative days could predict the 6- and 12-months outcome of KA.

**Methods**. Physiotherapists at two orthopaedic wards in Denmark were asked to predict the 6- and 12-months outcome of the KA patients they have treated post-operatively on a 0–10 scale (10 representing the best prognosis). At 6 and 12 months post-operatively the patients answered the Oxford Knee Score (OKS), EuroQol 5D-3L and Patient Acceptable Symptom State (PASS). Multivariable logistic regression analyses were performed to assess the prediction of PASS and treatment success. We assessed predictive performance by examining measures of calibration and discrimination.

**Results**. A total of 361 patients were included. The models for PASS and Treatment Success showed poor to acceptable discriminative values (OR between 1.47 and 1.92 and areas under the curves of 0.62–0.73), however the calibration plots indicated significant uncertainties in the prediction.

**Conclusion**. Physiotherapists prognoses of recovery after KA are associated with 6- and 12-months patient reported outcomes and satisfaction but have weak predictive value. This study suggests that physiotherapists' prognoses may be useful as an additional source of information when identifying patients in need of additional post-operative care.

Corresponding author
Marius Henriksen,
marius.henriksen@regionh.dk

## INTRODUCTION

Knee arthroplasty (KA) is considered a successful orthopaedic procedure to alleviate knee pain and disability in knee osteoarthritis (OA) and the demand for KA is large and growing worldwide (*Ackerman et al., 2019*; *Culliford et al., 2015*; *Nemes et al., 2015*). Despite being considered a generally safe and successful procedure, there is a significant proportion (20%) of patients who endure years of disability and dissatisfaction with their

postoperative function (*Bourne et al., 2010*; *Gunaratne et al., 2017*; *Kahlenberg et al., 2018*). Many of these patients do not undergo revision surgery, but all add to the society's burden of health care as clinicians and allied services strive to remedy their dissatisfaction.

While efforts have been made to predict the best candidates for KA (*Birch et al., 2019*; *Harbourne et al., 2019*; *Judge et al., 2012*) it has proven very difficult to identify robust pre-operative prognostic factors (*Gunaratne et al., 2017*). It is important to be able to identify inappropriate candidates pre-operatively, but until that is possible, efforts to improve post-operative rehabilitation and identify individuals that may need special attention post-operatively is equally important.

Physiotherapists play an important role in the post-operative treatment of patients that have received a KA. In general patients are mobilised very early, and rehabilitation is commenced during the first days after surgery (or even on the same day) while the patient is still hospitalised.

There are numerous factors that can predict the outcome of KA, of which most are either poor predictors or unsatisfactorily assessed scientifically (*Harmelink et al., 2017*). In daily clinical physiotherapy practice at the orthopaedic wards, previous experiences, knowledge, and personal interaction with a patient altogether result in the physiotherapist's intuition about the future course of a patient's recovery. Previously, it has been shown that physiotherapists' prognoses during the initial clinical encounter for the projected outcome of patients with low back and neck pain they treated, was associated with the actual clinical outcome (*Cook et al., 2015*). However, it is unknown if such intuition is a reliable predictor of the outcome of KA.

With this study we aimed to assess if physiotherapists treating inpatients at orthopaedic wards could predict the 6- and 12-months post-operative self-reported pain, functional status and health related quality of life of patients undergoing KA.

## MATERIALS & METHODS

The study is a prospective, pragmatic, longitudinal cohort study conducted from December 2016 to December 2019 with a 6- and 12-month follow-up. The study was conducted according to a prespecified protocol, which was pre-registered and submitted to the Health Research Ethics Committee of The Capital Region of Denmark (file number: 16039254). The committee deemed the study exempt from approval as the study only uses questionnaire data. Such studies can be implemented without approval from the Health Research Ethics Committee according to Danish legislation. The prespecified study protocol was pre-registered at http://www.clinicaltrials.gov (Identifier: NCT02982785) prior to data collection.

### Participants

Participants were recruited from the orthopaedic wards at Bispebjerg-Frederiksberg Hospital and Herlev-Gentofte Hospital in Copenhagen, Denmark. The inclusion criteria were: Primary total KA for knee OA; Age >18 years; Read and speak Danish and having an email-address. The exclusion criteria were: Revision surgery and cognitive or mental conditions precluding reliable answers to online questionnaires (determined either from

medical records or judged by the including physiotherapists). All participating patients signed an informed consent.

## Procedures

During the post-operative hospital stay, eligible participants were identified by the physiotherapists and invited to participate in the study. Upon signed informed consent, baseline data were collected using standardized forms. At discharge, physiotherapists delivering the initial in-patient physiotherapy and mobilization judged the participants' prognosis (see below).

Study data were collected and managed using REDCap electronic data capture tools hosted at The Capital Region of Denmark (*Harris et al., 2019*; *Harris et al., 2009*). REDCap (Research Electronic Data Capture) is a secure, web-based software platform designed to support data capture for research studies, providing (1) an intuitive interface for validated data capture; (2) audit trails for tracking data manipulation and export procedures; (3) automated export procedures for seamless data downloads to common statistical packages; and (4) procedures for data integration and interoperability with external sources. Via REDCap, individual internet-hyperlinks were emailed to the participants 6 and 12 months after discharge. The hyperlinks led to a secured webpage on which the patients answered questionnaires. The participant-submitted responses were automatically registered in a secured database. At both 6- and 12-months data collection points an email reminder was sent to participants if they did not answer the questionnaires within 7 days.

## Physiotherapists' prognoses

The physiotherapists estimated each patient's potential for a successful outcome after 6–12 months at discharge based on professional appraisal. The physiotherapists were asked to appraise all parts of their evaluation in their prognosis of each patient. The physiotherapists were instructed to score each patient on a 1–10 Likert scale (1 representing a very poor projected outcome, 10 representing an excellent projected outcome). The physiotherapists scored each patient following their complete encounter with the patient. This included the physiotherapist's assessment of the patient's resources (personal, material, social, etc.), personality, medical history, comorbidities, surgery reports, physical examination(s), in-patient physiotherapy treatment response(s), physiotherapeutic (re)assessments, and more. The prognosis was not disclosed to the participants.

## Outcomes

The primary outcome measure was the Oxford Knee Score (OKS). The Oxford Knee Score (OKS) is a 12-item Patient Reported Outcome questionnaire developed specifically to assess the patient's perspective of outcome following KA (*Dawson et al., 1998*). Standardized answer options are given (5 Likert boxes) and each question is assigned a score from 0 to 4. A total score is calculated that ranges from 0 and 48, with 48 indicating the best outcome. The OKS is short, practical, reliable, valid, and sensitive to clinically important changes over time (*Dawson et al., 1998*). The OKS scores were dichotomised using an established cut-off value for treatment success after TKA of 32.5 points (*Hamilton et al., 2018*). The

dichotomised OKS was labelled as 'Treatment Success' for scores above 32.5 at both 6- and 12-months follow-up.

In addition to the single OKS summary score, we also calculated the OKS pain and OKS function sub-scale scores that were standardized to a range from 0 (worst) to 100 (best).

Health outcome and quality of life was assessed using the European Quality of Life (EuroQoL) questionnaire (EQ-5D-3L). EQ-5D-3L is a standardized patient-reported instrument for use measuring health outcome and quality of life (*EuroQol, 1990*). EQ-5D-3L is designed for self-completion by respondents and is ideally suited for use in surveys.

The EQ-5D-3L consists of a descriptive system and a Visual Analogue scale (EQ-5D-VAS). The descriptive system comprises 5 dimensions (mobility, self-care, usual activities, pain/discomfort, and anxiety/depression). Standardized answer options are given (3 Likert boxes) and each question is assigned a score from 1 to 3. From the answers an EQ-5D-3L index score is calculated based on Danish normative equations. The index ranges from $-0.624$ (worst) to $1.000$ (best). The EQ-5D-VAS records the respondent's current self-rated health on a 10 cm Visual Analogue Scale (VAS) with endpoints labelled '*the best health you can imagine*' and '*the worst health you can imagine*'.

We also applied a single question regarding the Patient Acceptable Symptom State (PASS) (*Tubach et al., 2005*) to assess the patient's satisfaction with their state of symptoms at 6 and 12 months after surgery. PASS is assessed as a dichotomous outcome (yes/no) to the question: "*Considering your knee function, do you feel that your current state is satisfactory? With knee function you should take into account all activities you have during your daily life, Sport/Recreational activities, your level of pain and other symptoms, and your knee related QOL*".

## Statistical analyses

The 6- and 12-months follow-up data were analyzed separately. The associations between the physiotherapists' prognostic scores and the dichotomous PASS and Treatment Success variables were assessed using logistic regression analyses with the physiotherapists' prognostic scores as independent variable and PASS ('yes') and Treatment Success as dependent variables.

The predictive performance of the physiotherapists' prognostic scores was analysed using calibration and discrimination measures. Discrimination relates to the ability of the model to discriminate between patients who have answered 'yes' to the PASS question or achieved Treatment Success from those who have answered 'no' or did not achieve Treatment Success. These were evaluated by calculating the areas under the receiver operating characteristic curves (AUC) and Nagelkerke $R^2$ as an indication of explained variation. Calibration relates to the agreement between the projected and observed outcome and was evaluated by means of calibration plots, in which patients were classified by the predicted risk of the observed prognostic scores, supplemented by loess lines over the predicted probability range. The loess line of a prefect prediction model lies on the 45° slope for agreement with the observed outcome.

**Table 1  Age, sex and prognostic scores at baseline among all invited participants and the survey responders.**

| | All (n = 361) | Respondents at 6 months (n = 307) | Respondents at 12 months (n = 303) |
|---|---|---|---|
| Age | 69.2 (7.9) | 68.6 (7.9) | 68.8 (7.8) |
| Female, n (%) | 207 (57%) | 176 (57%) | 172 (57%) |
| PT Prognosis, n (%) | | | |
| 10 | 59 (16%) | 54 (18%) | 55 (18%) |
| 9 | 127 (35%) | 109 (35%) | 102 (34%) |
| 8 | 96 (27%) | 81 (26%) | 82 (27%) |
| 7 | 48 (13%) | 37 (12%) | 39 (13%) |
| 6 | 21 (6%) | 17 (6%) | 18 (6%) |
| 5 | 8 (2%) | 7 (2%) | 6 (2%) |
| 4 | 2 (1%) | 2 (1%) | 2 (1%) |

**Notes.**
PT, Physiotherapist.

The associations between the physiotherapists' prognostic scores and the continuous variables OKS, OKS-pain, OKS-function, EQ-5D-3L index and EQ-5D-VAS were assessed by univariate linear regression analyses with the physiotherapists' prognostic scores as independent variable and OKS, OKS-pain, OKS-function, EQ-5D-3L and EQ-5D-VAS as dependent variables. The associations were assessed via the estimated slopes (beta) and explained variance ($R^2$).

$P$-values of $<0.05$ were considered statistically significant.

## RESULTS

From December 2017 to December 2018 a total of 382 individuals (57% women) were invited to participate (Table 1). Of these 20 were not eligible and 1 did not have a prognostic score. Accordingly, 361 patients were included with a mean age of 69.2 years (SD 7.9). There were 42 patients who did not respond to any survey or reminders, 16 who responded to the 6-month but not the 12-month survey, and 12 who did not respond to the 6-months but responded to the 12-month survey (Fig. 1). Thus, at the 6-month follow-up 307 answered the survey and at 12 months 303 answered the survey.

The prognostic scores ranged from 4 to 10, with a mean of 8.3 (SD 1.2) and the median being 9 (Table 1). Summaries of the outcome variables are presented in Table 2.

At the 6-month follow-up, the incidence of positive answers ('yes') to the PASS was 240 (78%) and Treatment Success (OKS > 32.5) was 251 (82%). At the 12-month follow-up the incidence of positive PASS answer was 255 (85%) and of Treatment Success was 251 (83%).

A high prognostic score increased the odds of achieving PASS ('yes') at 6 months (OR = 1.47 (95% confidence interval 1.19–1.82)) and at 12 months (OR = 1.45 (95% CI 1.14–1.84)). Similarly, a high prognostic score increased the odds of Treatment Success at 6 months (OR = 1.62 (95% 1.29–2.03)) and at 12 months (OR = 1.92 (95% CI 1.51–2.46)).

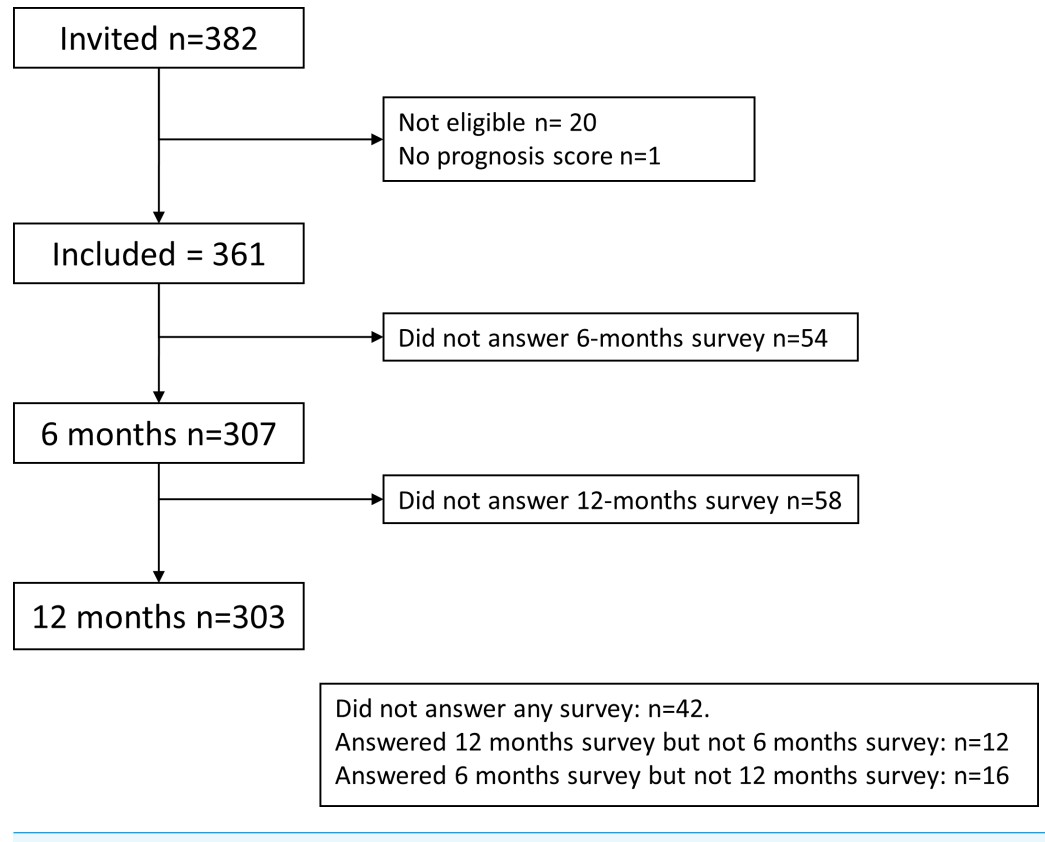

**Figure 1 Study flow chart.**

The performance of these models in terms of discrimination is presented in Table 3. The prediction of positive answer to the PASS and Treatment Success at 6 months showed statistically significant yet poor discrimination (AUC 0.62 and 0.67, respectively). At 12 months the prediction of positive answer to the PASS and Treatment Success showed poor to acceptable discrimination (AUC 0.64 and 0.73, respectively).

The calibration of the models is presented using calibration curves (Fig. 2) showing the performance of the models for PASS and Treatment Success at 6 and 12 months.

The results of the univariate regression analyses used to assess the prediction of the continuous outcomes are given in Table 4. The prognostic score was statistically significantly associated with all the dependent variables at both 6 and 12 months, although with limited explained variance ($R^2 < 0.12$; Table 4).

## DISCUSSION

In summary these findings support the hypothesis, that the prognostic scores of future outcomes made by physiotherapists attending patients undergoing KA in the first days postoperatively associate with the 6- and 12-month outcomes of KA. The discriminative performance can be considered poor to acceptable. However, the AUC and ROC curve analyses did not suggest a cut-off value that may be used to screen patients. Although

**Table 2  Group means and standard deviations (SD) in the different outcome measures at the 6- and 12-months follow-up.**

|  | Mean (SD) |
|---|---|
| OKS (0–48) | |
| 6 months | 37.6 (7.4) |
| 12 months | 39.7 (7.7) |
| OKS pain (0–100) | |
| 6 months | 80.2 (16.9) |
| 12 months | 84.7 (16.8) |
| OKS function (0–100) | |
| 6 months | 77.4 (15.9) |
| 12 months | 80.0 (16.8) |
| EQ-5D-3L index (−0.624–1.000) | |
| 6 months | 0.843 (0.145) |
| 12 months | 0.868 (0.153) |
| EQ-5D VAS (0–100) | |
| 6 months | 78.9 (16.7) |
| 12 months | 79.5 (18.3) |

Notes.
OKS, Oxford Knee Score; EQ-5D, EuroQoL 5 Dimensions; VAS, Visual Analog Scale.

**Table 3  Performance of prediction.**

|  | AUC (95% CI) | $R^2$ |
|---|---|---|
| PASS negative answer | | |
| 6 months | 0.62 (0.54–0.70) | 0.06 |
| 12 months | 0.64 (0.56–0.73) | 0.05 |
| No Treatment Success | | |
| 6 months | 0.67 (0.59–0.75) | 0.09 |
| 12 months | 0.73 (0.65–0.80) | 0.16 |

Notes.
AUC, Area Under the Curve; CI, Confidence Interval; PASS, Patient Acceptable Symptom State.

the calibration plots suggest a linear relationship between the predicted and observed probability of PASS and Treatment Success there are significant uncertainties as judged by the 95% confidence intervals. From Fig. 2 is can be seen that the uncertainties (width of the 95% CI) are related to the distribution of the predicted probabilities. This is likely due to the small number of patients with low prognostic scores (Table 1). The linear regressions show that the physiotherapists' prognostic scores are associated with the 6- and 12-months outcomes. Altogether the data suggest that the physiotherapists' prognoses associate with the outcomes, but the predictive values of the scores are not convincible. Thus, the physiotherapists' prognostic scores should be supported by other information and individual assessments (such as age, sex, body weight, post-operative pain and mobility, comorbidities etc.), when the future course of a patient is discussed and decisions about possible extra attention is taken. Further studies on defining patients that may not have successful outcomes of KA are needed.

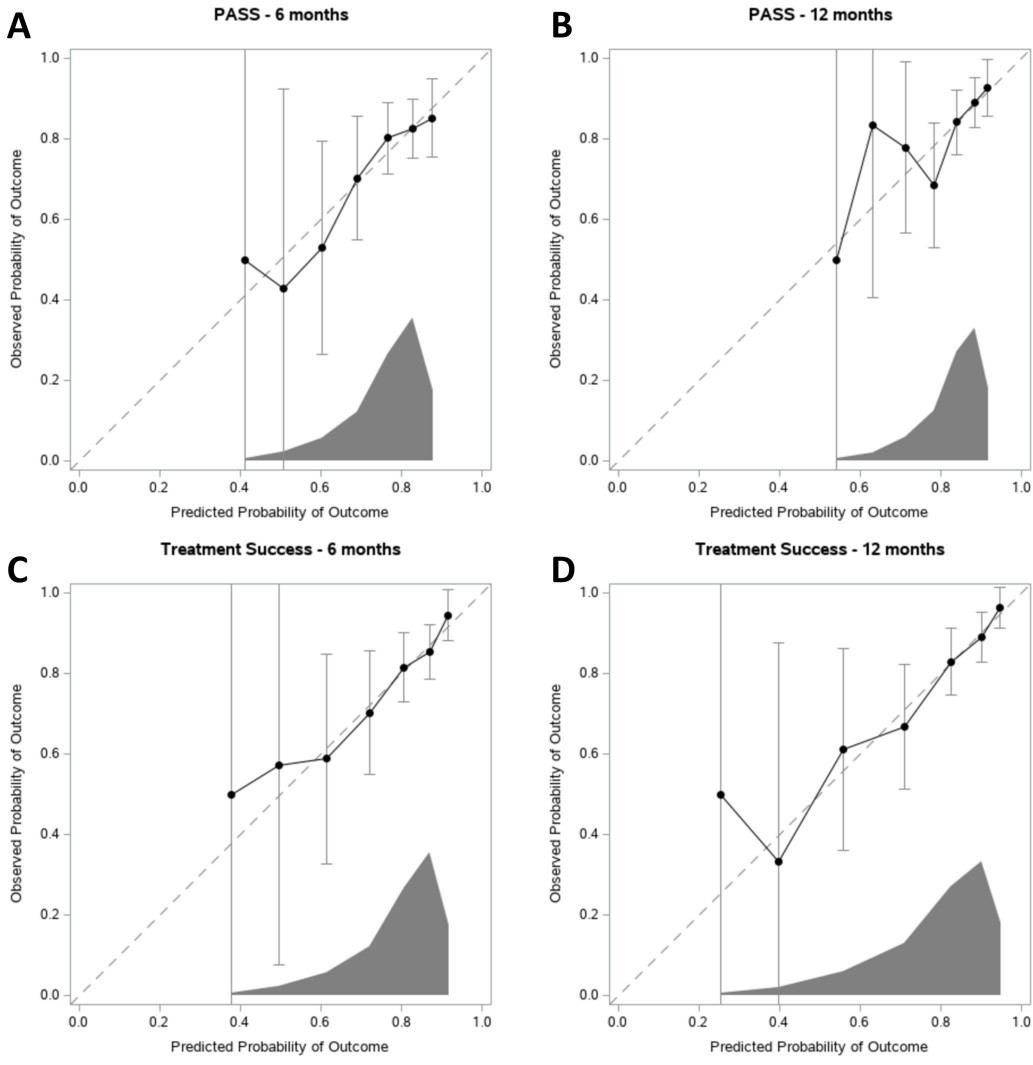

**Figure 2** **Calibration plots for PASS and Treatment Success at 6 and 12 months.** Predicted probability of achieving either (A–B) PASS or (C–D) Treatment Success is given on the *x*-axis, and the observed probability is given on the *y*-axis. The dashed diagonal line represents perfect agreement between the predicted and actual probability of PASS or Treatment Success. The dots represent the physiotherapists prognostic scores. Error bars are 95% confidence intervals. At the *x*-axis, the distribution of the predicted probabilities is shown.

The present study is the first on physiotherapists' prognoses of KA. Previously, it has been shown that physiotherapists' prognoses during the initial clinical encounter for the projected outcome of patients with low back and neck pain they treated, was associated with the actual clinical outcome (*Cook et al., 2015*).

A strength to the present study is the use of outcomes widely accepted for assessing outcomes of KA (*Conner-Spady et al., 2018*; *Dunbar et al., 2001*; *Garratt et al., 2004*; *Jin et al., 2019*). The total mean OKS score of the cohort, is comparable with the ones found in other studies (*Williams et al., 2013a*; *Williams et al., 2013b*), which support the external

**Table 4  Results of the univariate linear regression analyses with the physiotherapists' prognostic scores as predictor.**

| Dependent variable | 6 months | | | 12 months | | |
|---|---|---|---|---|---|---|
| | Slope (95% CI) | P | R$^2$ | Slope (95% CI) | P | R$^2$ |
| OKS | 1.9 (1.2–2.5) | <0.0001 | 0.10 | 2.1 (1.5–2.8) | <0.0001 | 0.12 |
| OKS Pain | 4.0 (2.5–5.4) | <0.0001 | 0.08 | 4.5 (3.0–5.9) | <0.0001 | 0.11 |
| OKS Function | 3.9 (2.5–5.2) | <0.0001 | 0.09 | 4.4 (2.9–5.8) | <0.0001 | 0.10 |
| EQ-5D-3L Index | 0.03 (0.02–0.05) | <0.0001 | 0.08 | 0.04 (0.03–0.06) | <0.0001 | 0.12 |
| EQ-5D VAS | 3.2 (1.8–4.7) | <0.0001 | 0.06 | 3.6 (2.0–5.2) | <0.0001 | 0.06 |

**Notes.**

OKS, Oxford Knee Score; EQ-5D, European Quality of Life 5 dimensions; VAS, Visual Analog Scale.

validity of the results. Also, the relatively large cohort provides some strength to the study. Further, the fact that the physiotherapists who made the prognoses were not involved in delivery of any post-operative rehabilitation is a significant strength as they could not influence the rehabilitation. Finally, it is a strength that the study was carried out on two independent hospitals in Denmark, which strengthen the generalizability of the results.

The study also has some limitations. Firstly, other types of information (such as pain intensity and functional disability, body weight, comorbidities etc.) from the pre-operative and first post-operative days could have been useful in identification of potentially modifiable factors (such as body weight, muscle strength, functional disability, adjustment in treatment of comorbidities) that associate with the prognoses. Such information could have helped identifying the underlying factors that the physiotherapists based their prognoses on and used to propose future interventions to mitigate negative outcomes. Further, the distribution of the prognostic scores were skewed, with few low scores, which probably reflects the general successfulness of KA and that the physiotherapists in general expect positive outcomes. Nevertheless, about 15% of the patients did not reach PASS or Treatment Success after 12 months and could be candidates for special post-operative attention . In this perspective, the physiotherapist prognosis may be used to inform the identification of patients who may benefit from more intensive post-operative care in order to enhance the outcome of KA. However, the present data suggest that such identification should not rely solely on the physiotherapists prognoses but should incorporate other sources of information. Also, the instruments used to assess treatment success and patient satisfaction (OKS and PASS) does not necessarily cover all aspects, as numerous factors determine these. An OKS score above 32.5 (treatment success) does not necessarily indicate success for the patient and the single yes/no PASS answer is not exhaustive. Several instruments to assess patient satisfaction after KA exist (*Kahlenberg et al., 2018*), yet there is no consensus on which instrument to use to assess treatment success or patient satisfaction. However, these instruments (OKS and PASS) are widely used in OA and KA clinical and research settings.

## CONCLUSIONS

In conclusion, physiotherapists prognoses of recovery after KA are associated with 6- and 12-months patient reported outcomes and satisfaction but have weak predictive value. This study suggests that physiotherapists' prognoses may be useful as an additional source of information when identifying patients in need of additional post-operative care.

### Funding

The authors received no funding for this work specifically. The Parker Institute is funded by grants from the Oak Foundation (grant no. OCAY-18-774-OFIL). There was no additional external funding received for this study. The funders had no role in study design, data collection and analysis, decision to publish, or preparation of the manuscript.

### Grant Disclosures

The following grant information was disclosed by the authors:
Parker Institute is funded by grants from the Oak Foundation: OCAY-18-774-OFIL.

### Competing Interests

Marius Henriksen is on the scientific advisory board of Thuasne.
Hiwa Mukriyani and Carsten Juhl declare that they have no competing interests.

### Author Contributions

- Marius Henriksen conceived and designed the experiments, performed the experiments, analyzed the data, prepared figures and/or tables, authored or reviewed drafts of the paper, and approved the final draft.
- Hiwa Mukriyani analyzed the data, prepared figures and/or tables, authored or reviewed drafts of the paper, and approved the final draft.
- Carsten Juhl conceived and designed the experiments, performed the experiments, prepared figures and/or tables, authored or reviewed drafts of the paper, and approved the final draft.

### Human Ethics

The following information was supplied relating to ethical approvals (i.e., approving body and any reference numbers):
The study protocol was submitted to the Health Research Ethics Committee of The Capital Region of Denmark (file number: 16039254). The committee deemed the study exempt from approval as the study only uses questionnaire data. Such studies can be implemented without approval from the Health Research Ethics Committee according to Danish legislation.

### Data Availability

Raw data are available as Supplemental File.

## Supplemental Information

Supplemental information for this article can be found online at http://dx.doi.org/10.7717/peerj.10838#supplemental-information.

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
