# Peer review of "Knee replacement outcome predicted by physiotherapists: a prospective cohort study"

_PeerJ, doi:10.7717/peerj.10838_

## Round 0.1 · original submission · Major Revisions

Dr Henriksen and colleagues. Thank you for your submission to PeerJ. Your manuscript has been reviewed and the Reviewers have noted changes required to your experimental design, the validity of findings and general comments. I belive you can address the issue raised by the Reviewers and would encourage you to amend your manuscript accordingly. I look forward to receiving your Response to Reviewers and your amended manuscript. Thank you again for your submission to PeerJ. A/Prof Mike Climstein

Reviewer 1 ·

Basic reporting

The article is mostly well-written to a professional standard, suggestions for rephrasing have been commented on in the manuscript.

There are several statements in the introduction and background that need to be rephrased, or better supported by referenced evidence, comments can be found in the manuscript.

Of particular note, could the author's please provide a rationale in the introduction that enhanced or intensive rehabilitation results in superior patient outcomes following KA.

Experimental design

Although the research question was well defined, it's clinical relevance was not. As per the comments above and in the manuscript, an evidence-based rationale for predicting those who will not reach treatment success at 12 months and providing intensive rehabilitation to lower that rate has not been established.

Methods: please define the exclusion criteria with sufficient detail to be reproducible. See comment in the manuscript regarding clarifying the process for determining cognitive status.

Validity of the findings

Please see specific comments within manuscript regarding the need for greater explanation for statements made, with the use of examples. Statements such as 'other types of information' or 'other sources of information' need examples to aid reader understanding.

Additional comments

The authors presented an interesting research question and the data obtained underwent robust statistical analysis.

However, the rationale for the relevance of this study has been poorly referenced, there are many statements that either need rephrasing or the addition of supporting references.

Secondly, the reporting of an AUC of 0.62 as being 'acceptable' needs referenced evidence to support this, as it may inflate the reporting of the performance of the model in the results.

Annotated reviews are not available for download in order to protect the identity of reviewers who chose to remain anonymous.

Reviewer 2 ·

Basic reporting

The report was written in a well structured and easy to read manner. There was appropriate terminology on the whole. The authors use the hunch in the introduction, and although I understand what they mean, I think that this term could be replaced.
In the introduction, and in some respect the discussion there was a lack of referencing. In particular, the introduction form lines 50-63 had no supporting evidence. There have been several studies into predicting outcome after knee arthroplasty that could have been referred to. I am unaware of studies in other field of medicine that may have looked into clinician predictive capabilities with outcome - this would have been good to incorporate to support the rationale (if such studies exist).
Table 2 was unclear - it presented mean and SD of outmeasures but the table title indicated least square means and group differences. This needs to be clarified. Figures require titles.
Raw data was shared.
Results and discussion reflect the hypothesis and the authors indicate that the predictive value should be used in conjunction with additional measures.

Experimental design

Original an interesting question that clinicians will find relevant. The research question was define and, as men toned above, could be supported further with evidence. There is a gap in this knowledge that the authors aim to address, but whether the results have further impact on defining groups that may not be successful is some thing for further studies and could be alluded to writhing the discussion.
Ethical approval was obtained and researchers state that the physiotherapists predicting the outcome had no input to the rehabilitation following discharge.
Methods - clarify if total KA. The determination of a "successful" KA is potentially a contentious one. A high OKS does not necessarily indicate success for the patient and the single PASS question Yes / No requires some further justification over other satisfaction scales.

Validity of the findings

The authors present the findings appropriately and do not negate any negative findings. The discussion points highlighted my initial concerns that the patients have a high predictive outcome by the physiotherapist and whether the cohort reflects that of all orthopaedic units. Therefore, comorbidity information would be beneficial for readers to understand the complexity of cases - more complex case are naturally more difficult to predict. A more heterogenous sample might strengthen the predictive modelling.
Conclusion represents study findings.

---

## Round 0.2 · accepted · Accept

Dr Henriksen and co-authors, thank you for your timely resubmission and meeting all of the issues identified by both reviewers. I am therefore pleased to inform you that your manuscript has been recommended to be accepted for publication.